# Pathway Phenotypes Underpinning Depression, Anxiety, and Chronic Fatigue Symptoms Due to Acute Rheumatoid Arthritis: A Precision Nomothetic Psychiatry Analysis

**DOI:** 10.3390/jpm12030476

**Published:** 2022-03-16

**Authors:** Hasan Najah Smesam, Hasan Abbas Qazmooz, Sinan Qayes Khayoon, Abbas F. Almulla, Hussein Kadhem Al-Hakeim, Michael Maes

**Affiliations:** 1Department of Chemistry, College of Science, University of Kufa, Kufa 540011, Iraq; hassann.hassan@uokufa.edu.iq (H.N.S.); headm2010@yahoo.com (H.K.A.-H.); 2Department of Ecology, College of Science, University of Kufa, Kufa 540011, Iraq; hasana.albuthabhak@uokufa.edu.iq; 3Department of Biology, College of Science, University of Kufa, Kufa 540011, Iraq; sinanq.almiarij@uokufa.edu.iq; 4Medical Laboratory Technology Department, College of Medical Technology, The Islamic University, Najaf 54001, Iraq; abbass.chem.almulla1991@gmail.com; 5Department of Psychiatry, Faculty of Medicine, Chulalongkorn University, Bangkok 10330, Thailand; 6Department of Psychiatry, Medical University of Plovdiv, 4002 Plovdiv, Bulgaria; 7IMPACT Strategic Research Centre, School of Medicine, Deakin University, P.O. Box 281, Geelong, VIC 3220, Australia

**Keywords:** precision nomothetic psychiatry, inflammation, neuro-immune, depression, chronic fatigue syndrome, anxiety

## Abstract

Rheumatoid arthritis (RA) is a chronic inflammatory and autoimmune disorder which affects the joints in the wrists, fingers, and knees. RA is often associated with depressive and anxiety symptoms as well as chronic fatigue syndrome (CFS)-like symptoms. This paper examines the association between depressive symptoms (measured with the Beck Depression Inventory, BDI), anxiety (Hamilton Anxiety Rating Scale, HAMA), CFS-like (Fibro-fatigue Scale) symptoms and immune–inflammatory, autoimmune, and endogenous opioid system (EOS) markers, and lactosylcer-amide (CD17) in RA. The serum biomarkers were assayed in 118 RA and 50 healthy controls. Results were analyzed using the new precision nomothetic psychiatry approach. We found significant correlations between the BDI, FF, and HAMA scores and severity of RA, as assessed with the DAS28-4, clinical and disease activity indices, the number of tender and swollen joints, and patient and evaluator global assessment scores. Partial least squares analysis showed that 69.7% of the variance in this common core underpinning psychopathology and RA symptoms was explained by immune–inflammatory pathways, rheumatoid factor, anti-citrullinated protein antibodies, CD17, and mu-opioid receptor levels. We constructed a new endophenotype class comprising patients with very high immune–inflammatory markers, CD17, RA, affective and CF-like symptoms, and tobacco use disorder. We extracted a reliable and replicable latent vector (pathway phenotype) from immune data, psychopathology, and RA-severity scales. Depression, anxiety, and CFS-like symptoms due to RA are manifestations of the phenome of RA and are mediated by the effects of the same immune–inflammatory, autoimmune, and other pathways that underpin the pathophysiology of RA.

## 1. Introduction

Rheumatoid arthritis (RA), a chronic autoimmune disease (CAID) that causes progressive damage to bone, cartilage, and joints in the hands and feet, may result in synovial membrane thickening and deformation and disabilities [1,2,3]. Inflammation of the synovial joints, which leads to joint pulverization and destruction, is associated with autoimmune responses to rheumatoid factor (RF) and anti-cyclic citrullinated peptide (anti-CCP), increased immune complexes, activated immune–inflammatory and nitro-oxidative pathways [4,5]. The inflammatory mediators that cause joint inflammation, including the proinflammatory cytokines interleukin (IL)-1β, IL-6, and tumor necrosis factor (TNF)-α, may spread systemically, leading to comorbid disease, including cardiovascular, pulmonary, kidney, and gastrointestinal disease [6,7]. IL-6 plays a significant role in the development of RA and is a key inducer of the acute phase response, resulting in elevated levels of CRP and ESR [8,9]. IL-6 is synthesized by endothelial cells and lymphocytes and synthesizes acute reactive proteins that can damage articular cartilage [10,11]. The granulocyte-macrophage colony-stimulating factor (GM-CSF) is an important factor promoting monocyte/macrophage survival in synovial liquid and membrane [12,13]. GM-CSF may aggravate arthritis in RA animal models, while a lack of functional GM-CSF in collagen-induced arthritis in mice shows protective activities [13,14]. IL-10, a negative immune regulatory cytokine, is often elevated in RA and is linked to increased RF and anti-CCP antibodies [15].

Some studies have suggested that inflammation and joint-destructive pathways in collagen-induced arthritis may be mediated by the Toll-Like-Receptor (TLR)4 and that TLR4 deficiency is associated with reduced autoimmune responses as indicated by lower IL-17 and anti-CCP levels [16,17]. TLRs are abundant in synovial tissues from RA patients, and TLR ligands may activate synovial fibroblasts and immune cell chemotaxis [18].

Some ceramides (lipid second messengers) mediate IL-1β and TNF-α signaling in RA synovial cells and cause apoptosis [19]. One of the ceramides, lactosylcer-amide (CD17), is involved in chemotaxis, phagocytosis, and superoxide generation, [20,21] as well as activating NADPH oxidase, resulting in the production of superoxide radicals (O^2^) [22] and inducible nitric oxide synthase (iNOS) [23]. There is also a report that RA is associated with changes in the endogenous opioid system (EOS), as evidenced by an inverse correlation between plasma β-endorphin levels, rheumatoid disease activity scores, and RA duration [24]. Acute inflammation with increased TNF-α and IL-1β levels causes leukocytes to release β-endorphins and mu (MOR) and kappa (KOR) opioid receptors [25,26]. These EOS compounds have antinociceptive and anti-inflammatory properties [26], suggesting that they may play a role in the pathophysiology of RA.

A significant proportion of RA patients exhibit symptoms of depression [27,28], anxiety [29,30,31], and chronic fatigue syndrome (CFS)-like symptoms [32,33]. According to one study, 67.5 percent of RA patients suffer from depression, with 60 percent suffering from moderate to severe depression [34]. CFS-like symptoms can be found in 40–70 percent of RA patients, with 41 percent experiencing severe fatigue [35]. The activated immune–inflammatory and nitro-oxidative pathways in that disorder may explain this high incidence of affective and CFS-like symptoms in RA [36]. Affective disorders including major depressive disorder (MDD) and generalized anxiety disorder, and CFS are neuro-immune and neuro-oxidative disorders characterized by immune activation (increased IFN-γ and IL-10), chronic low-grade inflammation (increased IL-1β, IL-6, and TNF-α), and increased nitro-oxidative stress (lipid peroxidation, protein oxidation) [37,38,39,40,41,42,43]. In RA, depressive symptoms are significantly associated with disease activity and immune activation markers, such as increased levels of IL-6 and IL-17 [34]. Inflammatory symptoms such as elevated CRP levels and tender and swollen joints are linked to increased fatigue [44]. A review reported that RA’s baseline IL-1 and interferon (IFN)-γ levels predict increased fatigue several weeks later [45].

TLR4 complex activation is thought to be important in affective disorders and CFS [46]. Ceramides, including lactosylcer-amide (CD17), are significantly higher in depressed patients than in controls [47,48,49]. Increased levels of CD17 are significantly associated with the severity of affective symptoms in people with type 2 diabetes mellitus (T2DM). [50]. Increased levels of β-endorphin and MOR have been found in major depression, and it is thought that they act as part of the negative immune-regulatory system (CIRS) [51,52,53]. Furthermore, we recently discovered an inverse relationship between the F-box/WD repeat-containing protein 7 (FBXW7) and affective symptoms, which was mediated by increased atherogenicity and insulin resistance [50]. F-box/WD repeat-containing protein 7 (FBXW7) is an E3 ubiquitin ligase that catalyzes the ubiquitin-mediated degradation of cyclin E [54]. FBXW7 controls neurogenesis by antagonizing c-Jun and Notch, two transcription factors required for neuronal differentiation and synaptic plasticity [55].

However, the precise relationships between affective and CFS-like symptoms, due to RA and the above-mentioned biomarkers, and severity of RA are not well defined. Hence, the purpose of this study was to examine if affective and CFS-like symptoms due to RA are associated with RA severity, autoimmune biomarkers (RF and anti-CCP), signs of immune activation (IL-6, CRP, GM-CSF, IL-10, soluble TLR4), and other biomarkers including CD17, FBXW7, and EOS biomarkers (β-endorphins, MOR, KOR, and endomorphin-2). The specific hypothesis is that affective and CFS symptoms due to RA are associated with increased immune–inflammatory biomarkers (IL-6, CRP, GM-CSF, TLR4, IL-10), increased RF and anti-CCP antibody levels, CD17, EOS compounds (KOR, MOR, and β-endorphin) but lowered FBXW7.

Moreover, in order to improve the existing RA disease model and to construct new endophenotype classes as well as pathway phenotypes, we applied the new precision nomothetic approach [43,56]. The latter comprises three steps: Step 1: improve the existing RA model (using supervised learning, namely Partial least Squares (PLS) path analysis) by enlarging the model with new indicators (e.g., CD17, EOS markers, affective and CFS-like symptoms); Step 2: discover new endophenotype classes (using unsupervised learning, including clustering techniques), and discover new pathway phenotypes (using supervised learning, including factor analysis); Step 3: compute personalized scores which shape an idiomatic profile for each individual [56].

## 2. Materials and Methods

### Participants

This case-control study was conducted in Al-Sader Teaching Hospital in Al-Najaf province, Iraq, from February 2021 till June 2019. The current study included 118 patients with RA flare-ups as well as 50 age- and gender-matched healthy controls. The “American College of Rheumatology” and the “European League Against Rheumatism” arthritis criteria were used to diagnose definite RA [57] based on synovitis in at least one joint and a score of 6 or more from scores in 4 domains, namely numbers and sites of affected joints, serologic abnormalities (RF and anti-CCP antibodies (ACPA)), acute-phase response (CRP and/or ESR), and symptom duration. Healthy controls were recruited from the same catchment area as family members or friends of staff and friends of patients. They were age and sex-matched with the patients. Inclusion criteria for patients and controls were the absence of any other systemic disease, including liver and heart disease and diabetes, any other axis-1 psychiatric disorder, and any neuroinflammatory or neurodegenerative disorder (including stroke, multiple sclerosis, Parkinson’s disorder, etc.). In addition, controls were excluded when they suffered from CFS, CF, Myalgic Encephalomyelitis, and RA or rheumatoid arthritis-like symptoms, or; as accessed 25 January 2022 showed a lifetime diagnosis of any axis-1 psychiatric disorder, including MDD and bipolar disorder dysthymia and protracted depression.

The Disease Activity Score (DAS28-CRP) was calculated using a calculator (https://www.das-score.nl/das28/DAScalculators/dasculators.html; as accessed 25 January 2022) based on the tender joint count (TJC), swollen joint count (SJC), CRP, and the patient’s general health measured using a visual analog scale (VAS). The online calculator (https://www.mdcalc.com/clinical-disease-activity-index-cdai-rheumatoid-arthritis; as accessed 25 January 2022) was employed to calculate the Clinical Disease Activity Index (CDAI). The simple disease activity index (SDAI) was computed using the online calculator (https://www.mdcalc.com/simple-disease-activity-index-sdai-rheumatoid-arthritis; as accessed 25 January 2022). The patient global assessment (PGA) score was a patient-reported outcome which was assessed as a single question scored from 0–10 (“How is your health overall”) [58]. The evaluator global assessment (EGA) assessed indexed disease activity (0–10 score) was evaluated by a senior rheumatologist [59].

The same day as the assessment of the severity of RA, a senior psychiatrist assessed the severity of depression using the Beck-Depression Inventory (BDI-II) score, [60] anxiety using the Hamilton Anxiety Rating Scale (HAMA), [61] and Fibromyalgia and Chronic Fatigue Syndrome Rating (FF) scale to measure the severity of CFS- and fibromyalgia-like symptoms [62]. An overall index of severity of psychopathology was computed as z total BDI-II score + z total FF score + z total HAMA score (z PP score). Using the median of the z PP score, we dichotomized the patient groups into two subgroups, namely patients with (RA + PP) and without (RA) increased z PP. We also computed two BDI-II subdomain scores, namely: (a) key depressive (Key_BDI) symptoms (sum of the items sadness, pessimism, loss of pleasure, loss of interest, past failure, guilty feelings, punishment feelings, self-dislike, self-criticalness, worthlessness); and (2) physio-somatic BDI-II (PS_BDI) symptoms (sum of the items loss of energy, changes in sleeping pattern, changes in appetite, concentration difficulties, tiredness or fatigue, loss of interest in sex). We also computed two HAMA subdomain scores, namely: (a) key anxiety (Key_HAMA) symptoms, namely sum of the HAMA items, anxious mood, tension, fears, and anxiety behavior at interview; and (2) physio-somatic HAMA (PS_HAMA) symptoms, namely sum of six symptoms, i.e., somatic sensory, and cardiovascular, respiratory, gastrointestinal, genitourinary and autonomic symptoms. Key FF (Key_FF) symptoms were computed as the sum of five symptoms: fatigue, autonomous and gastrointestinal symptoms, headache, and a flu-like malaise (thus excluding the cognitive, affective, and rheumatoid-like symptoms). The total physio-somatic score (z PS) was the sum of z PS_BDI score + z PS_HAMA + z key FF. We also computed a composite score reflecting “chronic fatigue” (z fatigue) as z BDII fatigue + z FF fatigue. There were no missing daa in any of the clinical data and biomarkers. Body mass index (BMI) was assessed ton he same day as the clinical interview as body weight in kg/length in m^2^. Tobacco use disorder (TUD) was diagnosed using DSM-IV-TR and DSM-5 criteria.

The ethics committee (IRB) of the University of Kufa, Iraq (243/2020), which complies with the International Guideline for Human Research Protection, approved the study as required by the Helsinki Declaration. Written informed consent was obtained from patients and controls before inclusion in the study.

## 3. Assays

After an overnight fast, all individuals had five milliliters of venous blood drawn and centrifuged for 15 min at 3000 rpm after full clotting at 37 °C. Sera were collected and kept at −80 °C until further analysis. Serum CRP was measured by a lateral flow immunoassay kit supplied by Shenzhen Lifotronic Technology^®^ Co. Ltd., Shenzhen, China. RF was measured using semi-quantitative kits provided by the Spinreact^®^ Co., Girona, Spain, based on the latex agglutination approach. A semi-quantitative anti-CCP assay was carried out using kits provided by Hotgen Biotech Co., Ltd., Beijing, China.

Commercial ELISA kits were used to measure serum β-endorphin, IL-6, CD17, and FBXW7 (Melsin^®^ Medical Co, Jilin, China), MOR, KOR, EM-2, and GM-CSF (MyBioSource^®^ Inc., San Diego, CA, USA), and IL-10 (Elabscience^®^, Wuhan, China). The sensitivities of the kits were for IL-6 and β-endorphin: 0.1 pg/mL, IL-10: 4.69 pg/mL: and MOR: 7.18 pg/mL, KOR: 1.0 ng/mL, EM-2: 0.33 pg/mL, CD17 and FBXW7: 0.1 ng/mL, and GM-CSF: 1.0 pg/mL. The intra-assay coefficients of variation (CV%) of all assays were less than 10%. The processes were carried out precisely as prescribed by the manufacturer, with no deviations. Sera were diluted at 1:4 to estimate CD17 levels. 

### 3.1. Statistical Analysis

The Chi-square test was used to assess associations between categorical variables, and the analysis of variance (ANOVA) test was used to compare scale variables across categories. We used Pearson’s product-moment correlation analysis and the point-biserial correlation analysis (the latter to assess the relationships between dichotomous variables (RF and anti-CCP) and continuous variables (e.g., clinical data)). Multivariate generalized linear model (GLM) analysis was utilized to check associations between diagnostic classes (RA subgroups and controls) and the measured biomarkers while controlling for age, sex, smoking and BMI entered as covariates. Subsequently, we used between-subject effects testing to check the associations between diagnosis and each significant biomarker. In this investigation, the partial eta-squared (η^2^) effect size was utilized. Automatic multiple regression analysis was used to predict dependent variables (clinical rating scale scores) based on biomarkers and demographic data while checking for R^2^ changes, multivariate normality (Cook’s distance and leverage), multicollinearity (using tolerance and VIF), and homoscedasticity (using White and modified Breusch-Pagan tests for homoscedasticity). We used an automatic stepwise (step-up) method with 0.05 *p*-to-enter and 0.06 *p*-to-remove. The results of these regression analyses were always bootstrapped with 5000 bootstrap samples, and the latter results are displayed if the results were not concordant. All tests are two-tailed, and a *p*-value of 0.05 was used to determine statistical significance. For statistical analysis, IBM SPSS Windows version 25, 2017, was used.

### 3.2. Precision Nomothetic Network Psychiatry

To optimize (enlarge) the RA model with new biomarkers and affective and CFS-like symptoms, we created a PLS model that included known auto-immune and immunological pathways, clinical indicators of the disease, and enriched this classical model with EOS and CD17 indicators, and BDI-II, HAMA and FF scores [56]. SmartPLS path analysis (SmartPLS) was used to determine the causal relationships between biomarkers and the RA phenome. Each variable was entered as either a single indicator or a latent vector derived from its reflective manifestations. When the outer and inner models complied with pre-specified quality data, complete PLS analysis was performed using 5000 bootstrap samples, namely: (a) the model fit SRMR is <0.08; (b) all latent vectors have good composite reliability (>0.7), Cronbach’s alpha (>0.7), and rho A (>0.8), with an average variance extracted >0.5; (c) all loadings on the latent vectors are >0.6 at *p* < 0.001; and (d) the latent vectors were not mis-specified as reflective models (tested with Confirmatory Tetrad Analysis, CTA). Complete PLS analysis was carried out with 5000 bootstrap samples and specific indirect, total indirect, and direct pathway coefficients (with exact *p*-value) were computed. The predictive performance was evaluated using PLS predict with 10-fold cross-validation Predicted-Oriented Segmentation analysis, Multi-Group Analysis, and Measurement Invariance Assessment were used to investigate compositional invariance [63].

In order to construct endophenotype classes, we performed cluster analysis on the latent variable scores extracted from all indicators in the PLS model that were associated (directly or indirectly) with the RA phenome. To this end, we used two-step and K-mean cluster analyses to discover clusters of individuals based on all significant features (performed using SPSS 28).

In order to construct pathway phenotypes we employed PLS path analysis and constructed a new latent vector which comprises adverse outcome pathways and symptomatome features and which is predicted by etiological factors that could not be combined with the latent vector. The new latent vector should comply with all quality criteria as discussed in the PLS section.

## 4. Results

### 4.1. Sociodemographic Data

Table 1 shows the sociodemographic and clinical data as well as PP rating scale scores in RA patients with RA + PP versus RA and the control group. We found no differences in sex, age, and BMI among the three groups. The RA + PP group showed higher CDAI and SDAI scores and TJCs and SJCs than the RA and control groups. The DAS28-4, total BDI-II. HAMA and FF scores increased from controls → RA → RA + PP group. There were no significant associations between TUD and either ACPA (χ^2^ = 0.01, df = 1, *p* = 0.916) or RF (χ^2^ = 0.07, df = 1, *p* = 0.796).

### 4.2. Multivariate GLM Analysis

Table 2 shows the associations between the diagnostic groups and all biomarkers combined in a multivariate GLM analysis while adjusting for age, sex, BMI, and smoking. A highly significant effect size of diagnosis (0.466) and a moderate effect size of TUD (0.295) were found. Sex, age, and BMI did not significantly affect the biomarker levels. Tests for between-subjects effects showed that diagnosis was highly significantly associated with CRP, IL-10, GM-CSF, IL-6, and TLR4. The association with CD17, KOR, FBXW7, and EP2 showed lower effect sizes. The model-generated estimated marginal means for the eleven biomarkers adjusted for age, BMI, sex, and smoking are shown in Table 3. The GM-CSF and CRP levels increased from control → RA → RA + PP. TLR4, FBXW7, CD17, IL-6, IL-10, EP2, and KOR were significantly higher in RA patients than controls. There were no significant differences in β-endorphins and MOR between the study groups.

### 4.3. Intercorrelation Matrix between Psychiatric Rating Scales and Biomarkers 

The results of the intercorrelation matrix between psychiatric rating scales and biomarkers are presented in Table 4. There is a strong significant correlation between the total BDI-II score and all biomarkers (except FBXW7). The total FF and z PP scores were significantly correlated with all biomarkers. The total HAMA score was significantly associated with all biomarkers except β-endorphins (results of Pearson’s product-moment correlation without *p*-correction). We have re-run the analysis in the restricted group of RA patients only and found that the correlations between all four rating scales and CRP and GM-CSF were significant (all at *p* < 0.01, *n* = 168). BDI was associated with TLR4 and IL-6 (at *p* < 0.05). MOR levels were positively associated with FF, HAMA, and z PP values (*p* < 0.05). EP2 was significantly correlated with BDI, FF, and z PP (*p* < 0.01).

### 4.4. Prediction of the Psychopathology Scores Using Biomarkers

Table 5 shows the results of multiple regression analyses with the PP scale scores as dependent variables and biomarkers as explanatory variables while allowing for the effects of age, sex, smoking, and BMI. Regression #1 shows that 57.9% of the variance in the z PP score was explained by the regression on CRP, GM-CSF, and TLR4. Figure 1 shows the partial regression plot of the z PP composite score on CRP (adjusted for age, sex, BMI, and IL-10). Regression #2 shows that 60.3% of the variance in the total BDI-II score was explained by CRP, GM-CSF, and TLR4. A considerable part of the variance in the total FF score (60.3%) was explained by CRP, GM-CSF, and EP2 (Regression #3). Regression #4 shows that 58.8% of the variance in the total HAMA score could be explained by the combined effects of CRP, GM-CSF, RF, anti-CCP, and TLR4. Regression #5 shows that 28.1% of the variance in Key_BDI was explained by the combined effects of CRP and GM-CSF. Regression #6 shows that 41.0% of the variance in the Key_HAMA symptom severity was explained by the combined effects of CRP, GM-CSF, and RF. 

### 4.5. Correlation of Psychiatric Rating Scale Scores and the Indices of Clinical Severity of RA 

The correlations between the psychiatric rating scale scores and the severity of RA in the entire study group (*n* = 168) and the restricted group of RA patients (*n* = 118) are presented in Table 6. The total BDI, FF, HAMA scores, and z PP scores were significantly and positively correlated with the DAS28-4, CDAI, SDAI, TJCs and SJCs, PGA, and EGA.

### 4.6. Prediction of the PP Scale Scores Using Biomarkers and RA Severity Scores

Table 7 shows the results of multiple regression analysis with the PP rating scale scores as dependent variables and the biomarkers and severity RA indices as explanatory variables while allowing for the effects of age, sex, BMI, and smoking. Regression #1 shows that 68.1% of the z PP composite score variance was explained by the variance in TJC and SJCs, CRP, GM-CSF (all positive), and IL-10 (inversely). Figure 2 shows the partial regression plot of the total BDI-II score on TJC (adjusted for EP2 and IL-10) and Figure 3 the partial regression of BDI-II scores on IL-10 (adjusted for EP2 and TJC). Regression #2 shows that 52.9% of the variance in the total BDI-II score was explained by IL-10 (negatively) and EP2, TJC, and SJC (all three positively). A considerable part of the variance in the total FF score (61.4%) was explained by CRP, GM-CSF, and TJC (all positively, seen in Regression #3). Regression #4 shows that 61.6% of the variance in HAMA was explained by the combined effects of CRP, GM-CSF, and TJC (all positively). Regression #5 shows that 49.6% of the variance in Key_BDI was explained by the combined effects of TJC and SJC (both positively) and IL-10 (inversely). Regression #6 shows that 42.5% of the variance in the Key_HAMA symptoms was explained by the combined effects of CRP and CDAI. The regression explained a significant part of Key_FF variance (41.3%) on CRP, GM-CSF, PGA, and EP2 (positively correlated), and IL-10 (negative) (Regression #7). Regression #8 shows that 57.9% of the variance in z PS could be explained by TJC, SJC, CRP, and EP2 (all positively), and IL-10 (inversely). A considerable part of the variance in z Fatigue (48.0%) was explained by CRP, GM-CSF, EP2, and TJC (positively), and IL-10 (negatively) (see Regression #9).

### 4.7. Precision Nomothetic Psychiatry Models 

#### PLS Analysis to Optimize the RA Disease Model

Figure 4 depicts the final PLS model, which investigated the causal pathways from autoimmunity and smoking to the phenome of RA. The latter included all RA severity measures (except DAS-28 and SDAI as they comprise CRP values) as well as all key PP and PS symptoms. The RA phenome was conceived as a latent vector (reflective model) derived from these 13 clinical scores. A latent vector (in a reflective model) was extracted from IL-6, IL-10, TLR4, CRP, and GM-CSF and reflected immune activation. In addition, EM2 and KOR were combined in a single latent formative vector. All other variables, including RF, anti-CCP, smoking, FBXW7, and CD17, were entered as single indicators. With SRMR = 0.048, the model’s overall fit was adequate. The phenome reflective latent vector’s construct reliability was adequate, with Cronbach = 0.954, rho A = 0.967, composite reliability = 0.960, and average variance extracted = 0.689. The loadings on this factor were greater than 0.666 at *p* < 0.0001 and blindfolding revealed that the construct cross-validated redundancy (0.457) was more than adequate. Furthermore, the immune activation factor construct reliabilities were adequate, with Cronbach = 0.817, rho_A = 0.836, composite reliability = 0.872, and average variance extracted = 0.578. At *p* < 0.0001, all loadings on this factor were greater than 0.666 and blindfolding revealed that the construct cross-validated redundancy (0.276) was adequate. CTA showed that the immune activation and phenome latent vectors were not mis-specified as reflective models. We discovered that 69.7 percent of the variance in the phenome latent vector was explained by immune activation, RF, anti-CCP, CD17, and MOR and that 50.2 percent of the variance in immune activation was explained by RF, anti-CCP, and smoking. Smoking and RF explained 18.1 percent of the variance in KOR + EP2. There were significant indirect effects of smoking on the phenome, mediated by immune activation (t = 2.97, *p* = 0.002) or MOR (t = 1.68, *p* = 0.046). There were also significant indirect effects of RF (t = 6.82, *p* < 0.001) and anti-CCP (t = 4.59, *p* < 0.001) on the RA phenome, which were both mediated by immune activation. All the endogenous construct indicators had positive Q^2^ Predict values, indicating that they outperformed the naive benchmark. The use of Predicted-Oriented Segmentation analysis in conjunction with Multi-Group Analysis and Measurement Invariance Assessment resulted in complete compositional invariance.

TJC: tender joint count, SJC: swollen joint count, PGA: patient global assessment, EGA: evaluator global assessment, CDAI: clinical disease activity Index, Key_FF: key symptoms of the Fibromyalgia and Chronic Fatigue Syndrome Rating (FF) scale, Key_BDI: key depressive symptoms of the Beck-Depression Inventory (BDI-II), KEY_HA: key anxiety symptoms of the Hamilton Anxiety Rating Scale (HAMA), PS_HA: physio-somatic symptoms of the HAMA, PS_BDI: physio-somatic symptoms of the BDI-II, z PS: integrated index of physio-somatic symptoms.

IL: interleukin, CRP: C-reactive protein, TLR4: Toll-Like Receptor 4, GM-CSF: granulocyte-macrophage colony-stimulating factor, FBXW7: F-box/WD repeat-containing protein 7, CD17: lactosylcer-amide, MOR: mu opioid receptor, KOR: kappa opioid receptor, EP2: endomorphin-2, RF: rheumatoid factor, anti-CCP: anti-citrullinated protein antibodies. 

### 4.8. Clustering Analysis to Discover New Endophenotype Clusters

K means clustering performed on all significant latent variable scores of all indicators, which were connected with the symptomatome (see Figure 4), revealed two factors with a good silhouette measure of cohesion and separation (0.62), namely the patient and control clusters. As such, this unsupervised technique optimized the current RA model and included more features of the illness as well as the psychopathology scores.

In order to discover new endophenotype classes within the patient group, we performed two-step cluster analysis with diagnosis (controls versus patients) as categorical variable and the CD17, MOR, immune, and phenome latent variable scores as continuous variables. This analysis revealed a three-cluster solution with a first cluster of normal controls, a second cluster with 95 RA cases, and a third cluster with 23 RA cases. The silhouette measure of cohesion and separation was good (0.52). Figure 5 shows the bar graph performed on the z scores of the model features. The immune, MOR and phenome (all *p* < 0.0001), but not CD17 (*p* = 0.561) latent variable scores were significantly different between RA cluster 1 and RA cluster 2. There were no significant differences in the prevalence of RF (*p* = 0.777) and anti-CCP (*p* = 0.344) antibodies between the latter clusters. The prevalence of TUD was significantly higher in RA cluster 2 (22/11) than in RA cluster 1 (90/5) (*p* < 0.001).

### 4.9. Construction of a Pathway Phenotype Using PLS Analysis

Figure 6 shows a second PLS path analysis with a new latent construct (combining symptoms, clinical and immune data), which is predicted by TUD, both autoimmune markers and IL-6 as input variables. The model overall fit was adequate (SRMR = 0.045) and the reflective latent vector construct reliability was adequate, with Cronbach alpha = 0.958, rho_A = 0.969, composite reliability = 0.963, and average variance extracted = 0.658. Moreover, all loadings on this latent construct were >0.6 with *p* < 0.0001. Blindfolding showed that the construct cross-validated redundancy (0.380) was adequate. Our results showed that 60.3 percent of the variance in the new latent construct was explained by smoking, anti-CCP, RF and IL-6.

RF: rheumatoid factor, anti-CCP: anti-citrullinated protein antibodies, smoking: tobacco use disorder.

CRP: C-reactive protein, GM-CSF: granulocyte-macrophage colony-stimulating factor, IL: interleukin, EGA: evaluator global assessment, PGA: patient global assessment, CDAI/SDAI: clinical/simple disease activity Index, SJC: swollen joint count, TJC: tender joint count, PS_HAMA: physio-somatic symptoms of the HAMA (Hamilton Anxiety Rating Scale), PS_BDI: physio-somatic symptoms of the Beck-Depression Inventory (BDI-II), Key_FF: key symptoms of the Fibromyalgia and Chronic Fatigue Syndrome Rating (FF) scale, Key_DEP: key depressive symptoms of the BDI-II, KEY_HAMA: key anxiety symptoms of the HAMA.

## 5. Discussion

### 5.1. Affective and CFS-Like Symptoms in RA

The study’s first major finding is the strong relationship between depressive, anxiety, CFS, and physio-somatic severity ratings and disease activity of RA as measured with PGA, EGA, CDAI, SDAI, DAS28, TJC, and SJC. Moreover, RA patients with increased ratings on the PP rating scales also showed increased scores on all these disease activity assessments compared with RA patients without increased PP ratings. These findings extend previous papers showing that CDAI scores were significantly associated with depression ratings. [64,65,66] In another study in RA, patients with depression showed higher ratings on TJCs, EGA, PGA, DAS-28, SDAI, and CDAI [65]. A cross-sectional study performed in The Kenyatta National Hospital showed that depression was accompanied by increased CDAI scores and lowered quality of life [67]. In another study, the CDAI score was significantly associated with self-reported depression and persistent depressive symptoms [64]. Treatment of RA with tocilizumab, an IL-6 antagonist, may reduce depression and anxiety ratings [68]. Moreover, depression in RA may cause increased disabilities and reduce a positive response to treatment and the likelihood of achieving remission [69,70,71,72]. Singh et al. (2014), reported that fatigue scores are significantly and positively associated with the DAS28, CDIA, TJC, SJC, PGA, and EGA [73]. Holten et al. (2021) found that a high number of RA patients suffer from fatigue (69%) and that this high prevalence decreased during treatment in association with PGA and SJC [74]. Many people with RA (up to 43%) show a somatization comorbidity phenotype (the PS symptoms measured in our study) and the response to treatment may be reduced in this group of patients [75]. All in all, these results show that RA disease activity is strongly associated with depression, anxiety, fatigue and physio-somatic symptom severity. Furthermore, there is some evidence that the relationships between depression and RA may be bidirectional because depression is also associated with an increased risk of RA and a more detrimental course of RA [76]. 

It should be stressed that we included only patients with depressive, anxiety and FF symptoms due to RA and excluded patients with a lifetime history of affective and anxiety disorders (and other axis I diagnoses), CFS or Myalgic Encephalomyelitis and, therefore, our results show that AR is associated with increased severity of “secondary” depression, anxiety and fatigue. Generally, medical patients with comorbid depression have more severe symptoms and higher medical costs than those who are not depressed due to their medical condition [77]. Future research should examine whether depression, anxiety, and CFS due to RA might affect the outcome of RA in terms of morbidity and mortality, including secondary cardio-vascular disorders, which are strongly associated with depression and CFS [78,79].

Most importantly, in the current study, we found that one common factor could be extracted from the diverse PP ratings and all RA disease activity indices and that this latent vector showed excellent psychometric properties, indicating that the three PP and RA disease activity scores are reflective manifestations of the same underlying construct or common core, which is the cause of the correlations among the PP and RA disease scores. It follows that increased PP ratings reflect the severity of RA and thus share the same pathophysiology as RA disease activity. Such findings exclude psychological theories that secondary PP symptoms should be ascribed to beliefs about the illness, the personal meaning attributed to stressors that accompany the illness, negative cognitions, or other folk psychology explanations [80]. Moreover, a large part (69.7%) of the variance in this common phenome core was explained by activated immune–inflammatory pathways, autoimmunity, CD17, and changes in the EOS system. In fact, the large effects sizes show that those signaling pathways are an essential part of the phenome of depression, anxiety, and FF due to RA patients. This will now be discussed in the following subsections.

### 5.2. Immune–Inflammatory Pathways 

The second major finding of this study is that different immune–inflammatory pathways were significantly associated with the common core and with the separate depression, anxiety, CFS, and physio-somatic scores. Previously, it was hypothesized that activated immune–inflammatory pathways may partly explain the occurrence of depression in RA [77]. Other authors also provided evidence that the bidirectional association between RA and depression may be explained by inflammation [76]. A recent review provided evidence that the relation between RA and depression can be explained by the effects of innate immune and molecular responses to inflammation [81]. Increased levels of proinflammatory cytokines, chemokines, type 1 interferons are considered to be the major culprits that may cause depression or explain the increased occurrence of depression in RA [81].

The present study showed that increased IL-6 and CRP in acute RA were significantly associated with PP symptoms due to RA. In a multivariate analysis study on RA severity, elevated serum IL-6 and CRP levels were associated with depression severity [34]. Moreover, patients with RA are at increased risk of developing depression, particularly if their disease activity scores and serum IL-6 levels are increased [34]. Increased levels of IL-6 and CRP are not only associated with major depression, [8,82] but also with RA, and in the latter illness, IL-6 is associated with increased CDAI, DAS28, and bone erosions [83]. The severity of CFS-like symptoms in schizophrenia, another neuroimmune disorder, is associated with immune–inflammatory markers, including increased plasma IL-6 [84]. Other authors suggested that higher levels of IL-6 are linked to both fatigue and pain and that this connection may contribute to the occurrence of depression [85]. Despite this, our findings show that affective symptoms and CFS belong to the same core, common to both PP and RA disease activity. This phenome core is highly associated with IL-6 and CRP, suggesting that all clinical manifestations share the same immune pathophysiology. Our research results imply that the idea that depression in RA is due to pain and fatigue should be rejected. 

The current study showed that increased GM-CSF, another cytokine that plays a key role in the immune–inflammatory response, is associated with PP symptoms due to RA. The GM-CSF levels are significantly increased in RA compared to healthy controls [86], while GM-CSF is also overexpressed in patients with depression [87]. Increased TLR4 mRNA and protein expression in PBMCs and synovial tissues are observed in RA patients [88]. Increased TLR4 signaling and mRNA TLR4 expression are both associated with depression [46,89]. Amaya-Amaya et al. suggested that interactions of pathogen-associated molecular patterns (bacterial or viral) with the TLR4 complex may be an initial inciting event in RA [1], which may subsequently cause activation of the innate immune system with increased levels of proinflammatory cytokines and reactive oxygen species [46]. In fact, the same mechanism has been proposed to initiate major depression and CFS [90,91].

In our study, increased levels of IL-10 were detected in RA, and they were positively associated with increased RA disease activity and PP assessments. Previously, it was shown that, in RA, IL-10 levels are positively correlated with RF, anti-CCP, and CRP [15,92], while there is also a well-established link between depression and elevated IL-10 [39,51,93]. The severity of CFS-like symptoms in schizophrenia is associated not only with increased IL-6 but also IL-10 levels [84]. Interestingly, we detected that, after considering the effects of inflammatory mediators (TJC, CRP, and GM-CSF) in regression analyses, IL-10 was inversely associated with depression, fatigue, physio-somatic, and overall PP ratings. These findings suggest that immune–inflammatory responses coupled with relative decrements in IL-10 may cause increased PP burden in RA. IL-10 has been shown to inhibit IL-6, TNF-α, and GM-CSF production from immune cells [94] and enhance B cell differentiation to cells secreting IgG, IgM, and IgA [95], resulting in increased RF and IgG-RF production by B cells in peripheral blood. Moreover, IL-10 is localized in the synovial membrane lining layer, the site of monocyte migration, and inhibits proinflammatory cytokines in RA [96,97]. In depression, IL-10 is one important component of the “compensatory immune-regulatory system” (CIRS) which tends to down-regulate the primary immune–inflammatory response [39,98].

Maes et al. (2011) reported that activated immune–inflammatory pathways and autoimmune and nitro-oxidative pathways may explain the occurrence of depression due to RA and another peripheral immune, autoimmune and neuroinflammatory disorders and that any bidirectional association may be explained by increased neuro-immune and neuro-oxidative signaling [36]. Indeed, depression is accompanied by autoimmune responses to oxidative-specific neoepitopes (including malondialdehyde and azelaic acid) and nitroso-adducts [36]. There is now evidence that O&NS pathways play a key role in RA, for example, by causing damage in the joints and inducing apoptotic processes in rheumatoid synovium and articular cartilage [99,100]. In RA, as well as in depression, there are multiple multidirectional interconnections among immune–inflammatory and O&NS pathways leading to a vicious circle between these two pathways. Moreover, such O&NS pathways also play a role in CFS and anxiety disorders [43,101], while similar autoimmune responses are also observed in depression and CFS [43].

### 5.3. Other Biomarkers of PP Due to RA

The current study found that PP symptoms were also associated with diverse alterations in the EOS and CD17 but only marginally with FBXW7. However, it should be stressed that the impact of these three biomarkers was modest compared to the major effect of immune–inflammatory pathways. Major depression is associated with elevated levels of MOR and β-endorphins [26], and the synovial membrane probably produces β-endorphin [102]. In recent years, substantial research points to the role of the EOS and their receptors in response to stress, mood regulation, and the pathophysiology of MDD [51]. Activation of immune–inflammatory pathways is likely responsible for circulating levels of β-endorphin and other EOS markers (such as MOR and KOR) during inflammatory conditions [103,104]. Successive expression of chemokines and adhesion molecules, primarily CXCR2 ligands, L and P selectins, α4/β2 integrins, and intercellular adhesion molecule 1 orchestrate the recruitment of opioid-containing leukocytes from the circulation to the site of inflammation [105,106,107]. EOS peptides and receptors are secreted during immune activation, and most of them act as CIRS compounds [26,51,103].

In the current study, we observed that lactosylcer-amide levels were significantly increased in RA compared to controls and were albeit marginally associated with the common phenome core extracted from the PP and RA disease activity data. Some ceramides, including CD17, are significantly elevated in depressed patients [47,48,49]. This is important because lactosylcer-amide plays a role in anxiety, neuroinflammation, reduced neurogenesis and neuronal survival, and neurodegenerative processes [108,109,110]. Moreover, inflammatory mediators, including TNF-α, activate the synthesis of lactosylcer-amide, which in turn activates ROS and α-type cytosolic phospholipase A_2_ (cPLA_2_α), which releases arachidonic acid, another inflammatory mediator [111]. Furthermore, lactosylcer-amide activates NADPH oxidase and iNOS and increases the production of superoxide radicals NO [22,23]. As such, the increased levels of lactosylcer-amide may participate in the immune–inflammatory pathophysiology of RA and the onset of depressive symptoms in RA. Moreover, in an antigen-induced RA model, acid sphingomyelinase, the lysosomal enzyme which hydrolyses membrane sphingomyelin to ceramide appears to be associated with joint swelling and increased levels of proinflammatory cytokines [22]. The lactosylcer-amide signaling pathways cause oxidative stress, inflammation, mitochondrial dysfunctions, and atherosclerosis [112] and, therefore, could play a role in RA-associated cardio-vascular disease. 

In contrast to the a priori hypothesis, we did not find lowered levels of FBXW7 in PP due to RA. Such changes could have contributed to the onset of depressive symptoms in RA. On the contrary, the FBXW7 levels showed very marginal positive associations with the PP scores.

### 5.4. Precision Nomothetic Psychiatry

As explained above, in the present study, we improved the existing RA disease model by constructing an enlarged causal, data-driven model which combines the building blocks of the disease (trigger or maintaining factors, adverse outcome pathways and the phenome), and incorporating new pathways (EOS and CD17) and affective and CFS-like symptoms. Moreover, we also uncovered a new endophenotype class with highly significantly increases in immune–inflammatory pathways, MOR levels, phenome severity, and TUD. Smoking is indeed a very important environmental factor that contributes to the pathophysiology of RA by increasing autoimmunity, inflammation, and oxidative stress [113,114]. Moreover, here we discovered a new pathway phenotype which consists of all clinical RA features, affective and CFS-like symptoms, and signs of immune activation. These results indicate that RA, affective, and CFS-like symptoms are manifestations of activated immune–inflammatory pathways which are triggered by autoimmunity, TUD and interkeukin-6 production. Finally, the latent variable scores and the scores on the new pathway phenotype shape an idiomatic profile or digital self of each RA patient and each control, thus opening the door towards personalized medicine [56].

## 6. Limitations

This study would have been more informative if we had measured other known biomarkers of depression including brain-derived-neurotrophic factor, which is frequently reduced in clinical depression [115]. Nevertheless, in RA increased levels of BDNF are detected which contribute to inflammatory responses [116]. As such, treatments that increase BDNF levels, including vagal nerve stimulation [117], may not be an asset to treat comorbid depression and RA. Second, since the EOS may be implicated in RA it would be worthwhile to trial the possible effects of a low dose naltrexone, which has anti-inflammatory effects [118], on alleviating pain symptoms [119]. Third, in the current study we recruited patients with RA flare-ups and a subgroup of the latter with comorbid depression symptoms. Due to those inclusion criteria, our results cannot be compared with epidemiologic studies reporting comorbidity prevalence in RA [120]. Fourth, due to our selection criteria we recruited RA subgroups with and without comorbid depression with high inflammatory burden. Such results differ from observational studies reporting on “not feeling well” one year after starting treatment despite achieving optimal RA control [121]. These patient-reported outcomes (PROs) and the high self-reported global assessment scores in remitted patients do not reflect subclinical inflammation [121,122]. Nevertheless, the consequences of flare-ups-associated inflammation—as for example increased oxidative stress and autoimmune responses directed to oxidatively modified neoepitopes—may underpin physio-somatic symptoms [123]. In addition, these PROs offer a different kind of information than that obtained using psychiatric interviews and ratings scales assessing anxiety and depression [56]. Moreover, clinical depression is a heterogeneous class and only patients with major symptoms show activated inflammatory pathways [56]. Fifth, our PLS pathway analysis, which was obtained using 5000 bootstrap samples, showed accurate quality data including predictive model assessments. Nevertheless, the models constructed here should be validated in independent patient groups, including in other countries.

## 7. Conclusions

Depression, anxiety, and CFS-like symptoms due to RA are reflective manifestations of the phenome of RA and are mediated by the effects of the same immune–inflammatory, autoimmune, EOS, and lactosylcer-amide pathways that underpin the pathophysiology of RA.

## Figures and Tables

**Figure 1 jpm-12-00476-f001:**
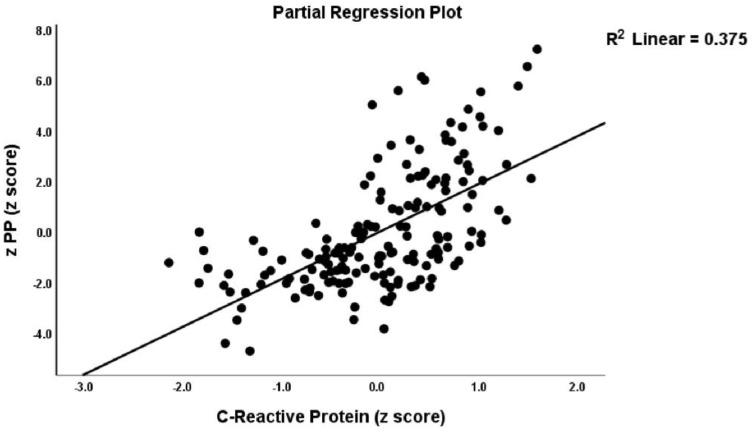
Partial regression plot of the integrated index of psychopathology (z PP) on C-Reactive Protein.

**Figure 2 jpm-12-00476-f002:**
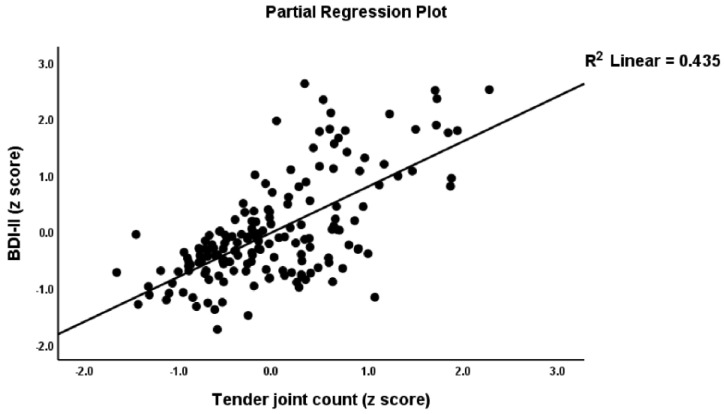
Partial regression plot of the total Depression Inventory (BDI-II) score on the tender joint count.

**Figure 3 jpm-12-00476-f003:**
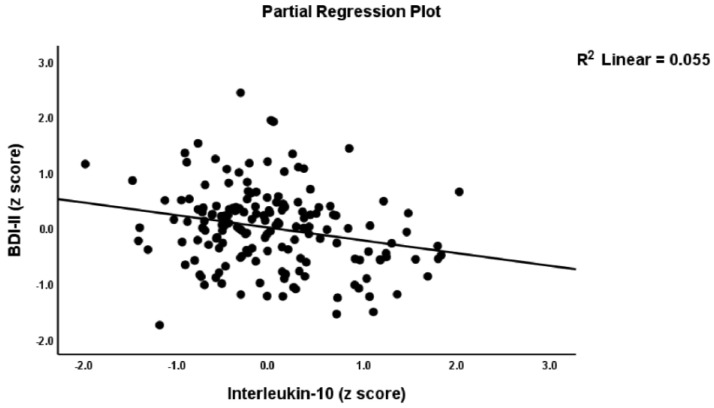
Partial regression plot of the total Depression Inventory (BDI-II) score on interleukin-10.

**Figure 4 jpm-12-00476-f004:**
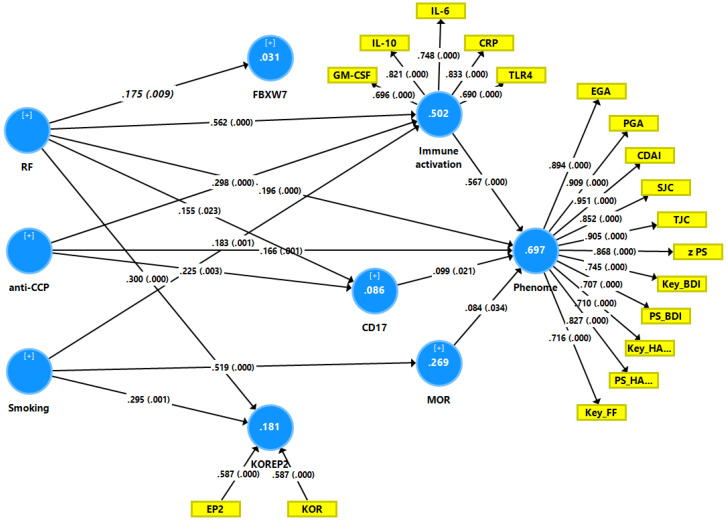
Results of Partial Least Squares (PLS) path analysis.

**Figure 5 jpm-12-00476-f005:**
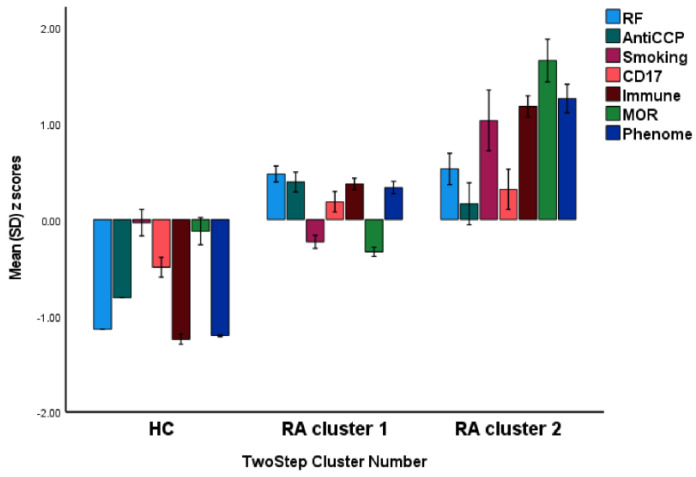
Results of cluster analysis performed to detect new endophenotype classes. Bar graph showing the z scores of the model features. RF: rheumatoid factor, smoking: tobacco use disorder, MOR: mu opioid receptor.

**Figure 6 jpm-12-00476-f006:**
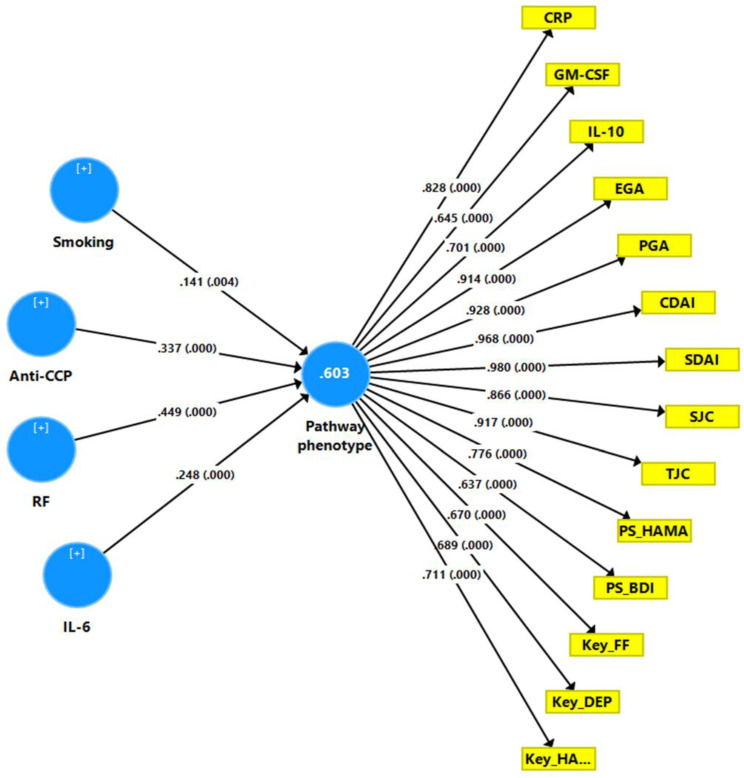
Results of a second Partial Least Squares path analysis, performed to construct a new pathway phenotype.

**Table 1 jpm-12-00476-t001:** Sociodemographic and clinical data in healthy controls (HC) and patients with rheumatoid arthritis divided into those with (RA + PP) and without (RA) increased psychopathology (PP).

Variables	HC ^A^ *n* = 50	RA ^B^ *n* = 59	RA + PP ^C^ *n* = 59	F/χ^2^	df	*p*
Age years	48.7 ± 4.6	49.8 ± 7.3	50.3 ± 6.8	0.90	2/165	0.407
Sex F/M	27/23	36/23	33/26	0.60	2	0.471
BMI kg/m^2^	27.44 ± 2.43	27.85 ± 2.65	27.60 ± 2.87	0.33	2/165	0.721
TUD No/Yes	44/6 ^C^	57/2 ^C^	45/14 ^A,B^	10.80	2	0.005
RF No/Yes	50/0 ^B,C^	14/45 ^A,C^	9/50 ^A,B^	93.50	2	<0.001
ACPA No/Yes	50/0 ^B,C^	27/32 ^A,C^	24/35 ^A,B^	47.54	2	<0.001
DAS28-4	1.61 ± 0 ^B,C^	4.50 ± 0.65 ^A,C^	5.26 ± 0.67 ^A,B^	650.49	2/165	<0.001
CDAI	0.0	16.7 ± 6.0 ^C^	22.9 ± 7.1 ^B^	KWT	-	<0.001
SDAI	0.0	20.1 ± 6.7 ^C^	28.3 ± 7.7 ^B^	KWT	-	<0.001
PGA	0.0	4.99 (1.97) ^C^	6.31 (2.23) ^B^	KWT	-	<0.001
EGA	0.0	4.36 (1.89) ^C^	5.72 (2.15) ^B^	KWT	-	<0.001
Tender joint count	0.0	4.9 ± 2.2 ^C^	7.2 ± 2.7 ^B^	KWT	-	<0.001
Swollen joint count	0.0	2.4 ± 1.6 ^C^	3.7 ± 1.4 ^B^	KWT	-	<0.001
Total BDI-II	8.9 ± 2.0 ^B,C^	11.3 ± 5.9 ^A,C^	27.5 ± 11.0 ^A,B^	103.62	2/165	<0.001
Total FF	5.9 ± 1.7 ^B,C^	12.8 ± 5.3 ^A,C^	24.5 ± 6.2 ^A,B^	203.44	2/165	<0.001
Total HAMA	6.0 ± 1.7 ^B,C^	13.2 ± 3.9 ^A,C^	23.1 ± 4.7 ^A,B^	288.14	2/165	<0.001

^A,B,C^: Pairwise comparison among group means, KWT: Kruskal-Wallis test. BMI: Body mass index, TUD: Tobacco use disorder, RF: Rheumatoid factor, ACPA: anti-citrullinated protein antibodies, DAS28-4(CRP): Disease Activity Score by four factors including CRP, CDAI: Clinical Disease Activity Index, SDAI: Simple Disease Activity Index, BDI: Beck Depression Inventory, FF: Fibro Fatigue Rating Scale, HAMA: Hamilton Anxiety Rating Scale, PGA: patient’s global assessment, and EGA: evaluator’s global assessment.

**Table 2 jpm-12-00476-t002:** Results of multivariate GLM analysis showing the associations between biomarkers and diagnosis while adjusting for background variables.

Type	Dependent Variables	Explanatory Variables	F	df	*p*	Partial η^2^
Multivariate	CRP, GM-CSF, TLR4, FBXW7, CD17, IL-6, IL-10, B-EN, MOR, EP2, KOR	HC/RA/RA + PP	11.50	22/290	<0.001	0.466
		Sex	0.68	11/144	0.753	0.050
		Age	1.08	11/144	0.378	0.077
		BMI	0.69	11/144	0.749	0.050
		TUD	5.47	11/144	<0.001	0.295
Tests for between-subject effects	CRP	HC/RA/RA + PP	234.54	2/154	<0.001	0.753
	IL-10	HC/RA/RA + PP	55.45	2/154	<0.001	0.419
	GM-CSF	HC/RA/RA + PP	33.49	2/154	<0.001	0.303
	IL-6	HC/RA/RA + PP	31.96	2/154	<0.001	0.293
	TLR4	HC/RA/RA + PP	21.19	2/154	<0.001	0.216
	CD17	HC/RA/RA + PP	7.71	2/154	0.001	0.091
	KOR	HC/RA/RA + PP	6.56	2/154	0.002	0.079
	FBXW7	HC/RA/RA + PP	6.16	2/154	0.003	0.074
	EP2	HC/RA/RA + PP	5.66	2/154	0.004	0.069
	MOR	HC/RA/RA + PP	1.07	2/154	0.346	0.014
	B-EN	HC/RA/RA + PP	0.67	2/154	0.515	0.009

HC/RA/RA + PP: healthy controls (HC) and patients with rheumatoid arthritis divided into those with (RA + PP) and those without (RA) increased psychopathology. CRP: C-reactive protein, GM-CSF: granulocyte-macrophage colony-stimulating factor, TLR4: Toll-like receptor 4, FBXW7: F-Box and WD Repeat Domain Containing 7, CD17: lactosylcer-amide, IL: interleukin, B-EN: β-endorphin, EP2: *endomorphin-2*, KOR: kappa opioid receptor, MOR: mu opioid receptor.

**Table 3 jpm-12-00476-t003:** Model-generated estimated marginal means values (SE) of the biomarkers in healthy controls (HC) and patients with rheumatoid arthritis (RA) divided into those with (RA + PP) and without (RA) increased psychopathology scores.

Biomarkers	HC ^A^ *n* = 50	RA ^B^ *n* = 59	RA + PP ^C^ *n* = 59
CRP mg/L	5.04 (2.06) ^B,C^	34.35 (1.92) ^A,C^	52.31 (1.93) ^A,B^
GM-CSF pg/mL	22.66 (6.98) ^B,C^	73.71 (6.51) ^A,C^	106.24 (6.53) ^A,B^
TLR4 ng/mL	4.01 (0.65) ^B,C^	9.60 (0.60) ^A^	9.33 (0.61) ^A^
FBXW7 ng/mL	16.08 (1.71) ^B,C^	25.09 (1.60) ^A^	22.38 (1.60) ^A^
CD17 ng/mL	1737.0 (207.6) ^B,C^	2777.4 (193.5) ^A^	2877.0 (194.0) ^A^
IL-6 pg/mL	6.8 (1.2) ^B,C^	11.5 (1.0) ^A^	13.0 (0.68) ^A^
IL-10 pg/mL	9.23 (0.98) ^B,C^	22.73 (0.91) ^A^	22.11 (0.91) ^A^
B-EN pg/mL	25.6 (2.6)	25.2 (2.3)	28.6 (2.4)
MOR pg/mL	5.12 (0.43)	5.32 (0.40)	5.93 (0.40)
EP2 pg/mL	383.0 (45.0) ^B,C^	522.8 (42.0) ^A^	588.8 (42.0) ^A^
KOR ng/mL	6.84 (0.74) ^B,C^	9.82 (0.69) ^A^	8.70 (0.70) ^A^

^A,B,C^: Pairwise comparisons among group means; CRP: C-reactive protein; GM-CSF: granulocyte-macrophage colony-stimulating factor, TLR4: Toll-Like receptor 4, FBXW7: F-Box and WD Repeat Domain Containing 7, CD17: lactosylcer-amide, IL: interleukin, B-EN: β-endorphin, EP2: *endomorphin-2*, KOR: kappa opioid receptor; MOR: mu opioid receptor.

**Table 4 jpm-12-00476-t004:** Intercorrelation matrix between psychopathology rating scales and biomarkers.

Biomarkers	Total BDI	Total FF	Total HAMA	z PP
CRP	0.562 ***	0.834 ***	0.736 ***	0.812 ***
GM-CSF	0.408 ***	0.616 ***	0.614 ***	0.634 ***
TLR4	0.304 ***/	0.421 ***	0.435 ***	0.449 ***
FBXW7	0.113	0.163 *	0.211 **	0.188 *
CD17	0.244 **	0.286 ***	0.321 ***	0.325 ***
IL-6	0.359 ***	0.471 ***	0.470 ***	0.496 ***
IL-10	0.342 ***	0.554 ***	0.559 ***	0.564 ***
B-EN	0.165 *	0.181 *	0.140	0.185 *
MOR	0.177 *	0.182 *	0.173 *	0.188 *
EP2	0.293 ***	0.339 ***	0.243 **	0.324 ***
KOR	0.188 *	0.252 **	0.270 ***	0.275 ***

* *p* < 0.05; ** *p* < 0.01, *** *p* < 0.001 (*n* = 168). BDI: Beck’s Depression Inventory, FF: Fibro Fatigue scale, Hamilton Anxiety Rating Scale (HAMA), z PP composite score reflecting overall psychopathology, CRP: C-reactive protein; GM-CSF: granulocyte-macrophage colony-stimulating factor; TLR4: Toll-like receptor 4; FBXW7: F-Box and WD Repeat Domain Containing 7; CD17: lactosylcer-amide; IL: interleukin; B-EN: Beta-endorphin; EP2: *endomorphin-2*; KOR: kappa opioid receptor; MOR: mu opioid receptor.

**Table 5 jpm-12-00476-t005:** Results of multiple regression analysis with psychiatry symptom domains as dependent variables.

Dependent Variables	Explanatory Variables	β	t	*p*	F _model_	df	*p*	R^2^
#1. z PP	**Model**				75.279	3/164	<0.001	0.579
	CRP	0.562	9.22	<0.001				
	GM-CSF	0.220	3.77	<0.001				
	TLR4	0.133	2.38	0.019				
#2. Total BDI	**Model**				25.844	3/164	<0.001	0.321
	CRP	0.374	4.83	<0.001				
	GM-CSF	0.181	2.45	0.015				
	TLR4	0.147	2.07	0.040				
#3. Total FF	**Model**				83.02	3/164	<0.001	0.603
	CRP	0.624	10.96	<0.001				
	GM-CSF	0.179	3.11	0.002				
	EP2	0.121	2.29	0.023				
#4. Total HAMA	**Model**				46.19	5/162	<0.001	0.588
	CRP	0.416	5.75	<0.001				
	GM-CSF	0.179	3.00	0.003				
	RF	0.172	2.64	0.009				
	Anti-CCP	0.129	2.25	0.026				
	TLR4	0.116	2.06	0.041				
#5. Key_BDI	**Model**				32.31	2/165	<0.001	0.281
	CRP	0.427	5.543	0.000				
	GM-CSF	0.183	2.422	0.016				
#6. Key_HAMA	**Model**				38.03	3/164	<0.001	0.410
	CRP	0.425	5.47	<0.001				
	GM-CSF	0.172	2.46	0.015				
	RF	0.160	2.12	0.035				
#7. Z PS	**Model**				63.19	2/164	<0.001	0.435
	CRP	0.447	6.67	<0.001				
	GM-CSF	0.315	4.71	<0.001				

BDI: Beck’s Depression Inventory, FF: FibroFatigue scale, HAMA: Hamilton Anxiety Rating Scale; Key_BDI: key depressive symptoms, Key_HAMA: key anxiety symptoms, z PS: composite score of all physio-somatic symptoms CRP: C-reactive protein, GM-CSF: granulocyte-macrophage colony-stimulating factor, TLR4: Toll-like receptor 4, IL: interleukin. EP2: *endomorphin-2*, RF: rheumatoid factor.

**Table 6 jpm-12-00476-t006:** Intercorrelation matrix of psychopathology rating scale scores and the indices of clinical severity of rheumatoid arthritis.

Parameter	Total BDI	Total FF	Total HAMA	z PP
DAS28-4	0.637/0.609	0.782/0.545	0.779/0.497	0.838/0.632
CDAI	0.596/0.545	0.733/0.422	0.751/0.421	0.799/0.533
SDAI	0.619/0.592	0.777/0.537	0.771/0.484	0.829/0.617
Tender joint count	0.625/0.641	0.711/0.368	0.752/0.415	0.799/0.553
Swollen joint count	0.590/0.449	0.711/0.332	0.726/0.322	0.771/0.425
PGA	0.511/0.373	0.720/0.372	0.723/0.335	0.762/0.411
EGA	0.494/0.325	0.713/0.340	0.719/0.322	0.754/0.374

Nominator: correlations are computed in patients and controls combined (*n* = 168); denominator: correlations are computed in patients only (*n* = 118). All correlations at *p* < 0.001. BDI: Beck Depression Inventory, FF: Fibro-Fatigue scale, HAMA: Hamilton Anxiety Rating Scale, z PP composite score reflecting overall psychopathology, DAS28-4: Disease Activity Score by four factors including C-Reactive Protein, CDAI: Clinical Disease Activity Index, SDAI: Simple Disease Activity Index.

**Table 7 jpm-12-00476-t007:** Results of multiple regression analysis with psychiatric symptom domains as dependent variables.

Dependent Variables	Explanatory Variables	β	t	*p*	F _model_	df	*p*	R^2^
#1. z PP	**Model**				69.27	5/162	<0.001	0.681
	Tender joint count	0.445	5.61	<0.001				
	CRP	0.338	4.94	<0.001				
	GM-CSF	0.113	2.08	0.039				
	IL-10	−0.165	−2.64	0.009				
	Swollen joint count	0.166	2.22	0.028				
#2. Total BDI	**Model**				43.84	4/156	<0.001	0.529
	Tender joint count	0.673	7.60	<0.001				
	IL-10	−0.267	−3.52	0.001				
	Swollen joint count	0.189	2.17	0.031				
	EP2	0.132	2.16	0.032				
#3. Total FF	**Model**				86.83	3/164	<0.001	0.614
	CRP	0.508	7.21	<0.001				
	Tender joint count	0.231	3.15	0.002				
	GM-CSF	0.150	2.56	0.011				
#4. Total HAMA	**Model**				87.87	3/164	<0.001	0.616
	Tender joint count	0.404	5.52	<0.001				
	CRP	0.353	5.03	<0.001				
	GM-CSF	0.136	2.33	0.021				
#5. Key_BDI	**Model**				53.72	3/164	<0.001	0.496
	Tender joint count	0.674	7.34	<0.001				
	IL-10	−0.219	−2.90	0.004				
	Swollen joint count	0.193	2.14	0.034				
#6. Key_HAMA	**Model**				60.99	2/165	<0.001	0.425
	CDAI	0.389	4.25	<0.001				
	CRP	0.305	3.34	0.001				
#7. Key_FF	**Model**				22.63	5/161	<0.001	0.413
	CRP	0.327	3.45	0.001				
	GM-CSF	0.240	3.27	0.001				
	PGA	0.273	2.78	0.006				
	IL-10	−0.229	−2.67	0.008				
	EP2	0.143	2.11	0.036				
#8. z PS	**Model**				36.64	6/160	<0.001	0.579
	Tender joint count	0.449	4.82	<0.001				
	GM-CSF	0.181	2.83	0.005				
	IL-10	−0.305	−4.09	<0.001				
	CRP	0.215	2.72	0.007				
	Swollen joint count	0.188	2.16	0.032				
	EP2	0.122	2.13	0.035				
#.9 z Fatigue	**Model**				29.88	5/162	<0.001	0.480
	Tender joint count	0.404	4.46	<0.001				
	CRP	0.299	3.49	0.001				
	EP2	0.204	3.21	0.002				
	IL-10	−0.267	−3.30	0.001				
	GM-CSF	0.170	2.43	0.016				

BDI: Beck Depression Inventory, FF: Fibro Fatigue Rating Scale, HAMA: Hamilton Anxiety Rating Scale, Key_BDI: key depressive symptoms, Key_HAMA: key anxiety symptoms, z PP: a composite score reflecting overall psychopathology, z PS: composite score of all physio-somatic symptoms, z Fatigue: a composite score reflecting fatigue CRP: C-reactive protein; GM-CSF: granulocyte-macrophage colony-stimulating factor; IL: interleukin; EP2: *endomorphin-2*.

## Data Availability

The dataset generated during and/or analyzed during the current study will be available from MM upon reasonable request and once the authors have fully exploited the dataset.

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
