# Peer review of "Pathway Phenotypes Underpinning Depression, Anxiety, and Chronic Fatigue Symptoms Due to Acute Rheumatoid Arthritis: A Precision Nomothetic Psychiatry Analysis"

_jpm, 2022, doi:10.3390/jpm12030476_

Round 1

Reviewer 1 Report

This is a thorough and interesting review of the involvement of the immune system in generating depression symptoms in patients with rheumatoid arthritis.My only criticism is the need to integrate with findings of other biomarkers of depression such as plasma levels of brain derived neurotrophic factor and the possibility that vagal stimulation might be particularly appropriate treatment of RA as it is of depression that is known to stimulate BDNF and can reduce markers of inflammation perhaps via its ability to simulate the release of oxytocin which is known to counter the expression of inflammatory markers induced by endotoxin.With respect to markers of endorphin release is there any evidence of worsening symptoms following treatment with naloxone?

Author Response

This study would have been more informative if we had measured other known biomarkers of depression including brain-derived-neurotrophic factor, which is frequently reduced in clinical depression [115]. Nevertheless, in RA increased levels of BDNF are detected which contribute to inflammatory responses [116]. As such, treatments that increase BDNF levels, including vagal nerve stimulation [117], may not be an asset to treat comorbid depression and RA. Second, since the EOS may be implicated in RA it would be worthwhile to trial the possible effects of a low dose naltrexone, which has antiinflammatory effects [118], on alleviating pain symptoms [119].

Reviewer 2 Report

Dear authors,

Thank you for this interesting paper about an indepth analysis of psychopathology of patients with RA. I have some comments which should be addressed in a limitation section that I can not seem to find in the paper.

  1. Population selection. How representative is this "acute" RA population? Firstly, authors use some sort of ACR diagnosis criteria to define RA. Do I have to interpret this that this acute RA population are newly diagnosed patients with RA or is this a severe RA population without taking into account disease duration? Moreover, the healthy control seems really healthy in psychopatholoy. Recent studies such as (https://www.ncbi.nlm.nih.gov/pmc/articles/PMC8220534/pdf/rmdopen-2021-001671.pdf) show that in prevalent RA case control studies, comorbidities are present over 50% in both case and controls and that depression rates (around 12-13%) are similar. Authors should reflect how generalisable their results are in this context, also considering the geography of their sample.
  2. Authors here put forward that depression is mostly immune-inflammatory driven. I have two reflections on this. Depression seems ubiquitous in modern society. Only attributing it to such pathways seems going a step too far. Moreover, even in patients in sustained remission PRO scores can be high. There is a whole work discussing why patients report "bad" PRO scores while patients are clinically ok. (See https://pubmed.ncbi.nlm.nih.gov/32371432/, https://pubmed.ncbi.nlm.nih.gov/34166795/ and many more). Authors only looked upon a very active high in inflammation population which causes a selection towards the inflammatory theory.
  3. I`m a bit surprised that smoking and ACPA are so different in the models. These are highly correlated.
  4. Many models and tests have been used in a population of modest sample size. This should be atleast mentioned in the limitations.
  5. Were there missing data? How was this handled?
  6. Was the ML method also performed in SPSS?
  7. For ML methods i expect a test and validation cohort. This was not done. How can the authors convince me of the robustness of their results?

Author Response

These remarks are now addressed in the new limitations section. It reads:

  1. Third, in the current study we recruited patients with RA flare-ups and a subgroup of the latter with comorbid depression symptoms. Due to those selection criteria focusing on comorbid RA and depression, our results cannot be compared with epidemiologic studies reporting comorbidity prevalences in RA [120].

I did not address  “Moreover, the healthy control seems really healthy in psychopathalogy”, which is a contradiction in terms. In my opinion, normal controls are normal and thus free of psychopathology. Maybe they have some PROs, which many psychologists and psychiatrists (but not scientists who examine major medical illness like major depression) consider psychopathology.

  1. Fourth, due to our selection criteria we recruited RA subgroups with and without comorbid depression with high inflammatory burden. Such results differ from observational studies reporting on “not feeling well” one year after starting treatment despite achieving optimal RA control [121]. These patient-reported outcomes (PROs) and the high self-reported global assessment scores in remitted patients do not reflect subclinical inflammation [121,122]. Nevertheless, the consequences of flare-ups-associated inflammation - as for example increased oxidative stress and autoimmune responses directed to oxidatively modified neoepitopes - may underpin physiosomatic symptoms [123]. In addition, these PROs offer a different kind of information than that obtained using psychiatric interviews and ratings scales assessing anxiety and depression [56]. Moreover, clinical depression is a heterogeneous class and only patients with major symptoms show activated inflammatory pathways [56].

  1. There no associations between TUD (smoking) and ACPA and RF. Now added to the Results section:

There were no significant associations between TUD and either ACPA (χ2=0.01, df=1, p=0.916) or RF (χ2=0.07, df=1, p=0.796).

Nevertheless, our analyses show that TUD is a very important player in RA (if you ask me: via inflammation and especially oxidative stress).

  1. Listed as a limitation: Fifth, our PLS pathway analysis, which was obtained using 5.000 bootstrap samples, showed accurate quality data including predictive model assessments. Nevertheless, the models constructed here should be validated in independent patient groups, including in other countries.

Modest sample size?? This study is performed at a power > 08 (probably 0.95). Tis is not an epidemiologic study, but a biomarker study.

  1. Added to the Statistics section:

There were no missing daa in any of the clinical data and biomarkers as well.

  1. We now desribe in the Statistics that we used Smart PLS and SPSS28 to perform these tests (see Satistics)

  1. see remark 4

Round 2

Reviewer 2 Report

Dear authors,

thank you for the answers to my comments although the tone of the answer seems out of place. Please remember reviewers review papers outside their working hours for no money. A bit of respect is the least I expect. Moreover, being angry at comments does not display any academic behaviour. i would recommend the authors to change the tone and words of their response in the future.

  1. Patients were included if they had a flare-up. How was this defined?
  2. By selecting only patients with a flare-up (which implies high inflammation markers I suppose?), authors perhaps induce a selection bias if you want to look at the mechanistics of the discussed psychopathologies.
  3. If you have 118 RA cases and 50 matched controls, this implies some cases remain unmatched. Why was such an approach chosen? I agree that is may sound contra intuitive that the healthy control population seemed too healthy, yet you would assume that age and gender matched controls also have some comorbidities present.
  4. Authors state that 118 RA cases give a lot of power to answer all questions proposed here. I`m curious how this was calculated. Could authors detail this?
  5. Interesting that smoking and ACPA are completely unrelated. This may arise as authors use TUD, which seems perhaps a degree worse than just being a proxy of smoking behavior. Why was TUD chosen and not smoking (yes/no) or packyears?
  6. No missing data was present. Does it imply that patients with some missing data were omitted from inclusion in the study?